# Early Biomarkers for Detecting Subclinical Exposure to Fumonisin B1, Deoxynivalenol, and Zearalenone in Broiler Chickens

**DOI:** 10.3390/toxins17010001

**Published:** 2024-12-24

**Authors:** Laharika Kappari, Todd J. Applegate, Anthony E. Glenn, Abhijeet Bakre, Revathi Shanmugasundaram

**Affiliations:** 1Department of Poultry Science, University of Georgia, Athens, GA 30602, USA; 2Toxicology and Mycotoxin Research Unit, USDA-ARS, Athens, GA 30605, USA; 3Exotic and Emerging Avian Viral Diseases Research, USDA-ARS, Athens, GA 30605, USA

**Keywords:** mycotoxin, biomarker, broiler chickens

## Abstract

Identifying biomarkers of mycotoxin effects in chickens will provide an opportunity for early intervention to reduce the impact of mycotoxicosis. This study aimed to identify whether serum enzyme concentrations, gut integrity, and liver miRNAs can be potential biomarkers for fumonisin B1 (FB1), deoxynivalenol (DON), and zearalenone (ZEA) toxicity in broiler birds as early as 14 days after exposure. A total of 720 male broiler chicks were distributed to six treatment groups: T1: control group (basal diet), T2 (2 FB1 + 2.5 DON + 0.9 ZEA), T3 (5 FB1 + 0.4 DON + 0.1 ZEA), T4 (9 FB1 + 3.5 DON + 0.7 ZEA), T5 (17 FB1 + 1.0 DON + 0.2 ZEA), and T6 (21 FB1 + 3.0 DON + 1.0 ZEA), all in mg/kg diet. On d14, there were no significant differences in the body weight gain (BWG) of mycotoxin treatment groups when compared to the control (*p* > 0.05), whereas on d21, T6 birds showed significantly reduced BWG compared to the control (*p* < 0.05). On d14, birds in T6 showed significant upregulation of liver miRNAs, gga-let-7a-5p (14.17-fold), gga-miR-9-5p (7.05-fold), gga-miR-217-5p (16.87-fold), gga-miR-133a-3p (7.41-fold), and gga-miR-215-5p (6.93-fold) (*p* < 0.05) and elevated serum fluorescein isothiocyanate-dextran (FITC-d) concentrations, aspartate aminotransferase (AST), and creatine kinase (CK) levels compared to the control (*p* < 0.05). On d21, T2 to T6 birds exhibited reduced serum phosphorus, glucose, and potassium, while total protein, FITC-d, AST, and CK levels increased compared to control (*p* < 0.05). These findings suggest that serum FITC-d, AST, CK, and liver miRNAs could serve as biomarkers for detecting mycotoxin exposure in broiler chickens.

## 1. Introduction

Broiler chicken diets primarily consist of cereals, which are often contaminated with multiple mycotoxins [1]. According to the Food and Agriculture Organization (FAO), above 25% of the global food crops are contaminated with mycotoxins [2], causing a USD 5 billion annual economic loss for North America [3]. The fungi *Fusarium verticillioides* and *Fusarium proliferatum* are prominent pathogens of corn and produce fumonisins (FB1, FB2, and FB3), while *Fusarium graminearum* and *Fusarium culmorum* (and other species) infect corn, wheat, and other grains and produce the mycotoxins deoxynivalenol (DON) and zearalenone (ZEA) [4,5].

According to the Food and Drug Administration (FDA), the maximum permissible levels for FB1 and DON in the finished feed for broiler chickens are 50 and 5 mg/kg, respectively [6], and no guidelines have been established for ZEA in growing chickens. Additionally, according to the European Union (EU), the maximum permissible levels for fumonisins (FB1 + FB2), DON, and ZEA in poultry feed are 20 mg/kg, 5 mg/kg, and 0.25 mg/kg, respectively [7]. In North American commercial poultry settings, 92% of the finished diet contains more than one mycotoxin, according to the BIOMIN mycotoxin survey, 2023 [8]. Among these, fumonisins (FBs) and DON are the most prevalent mycotoxins detected in 70% of the tested corn samples. The average concentration of FBs was 4.2 mg/kg, with a maximum of 83.3 mg/kg, while DON averaged 1.4 mg/kg, with a maximum concentration of 11.3 mg/kg detected in corn samples [8]. This co-occurrence of multiple mycotoxins in poultry feed [9], even at subclinical doses, which are well below FDA regulations of 50 mg/kg diet of FB and 5 mg/kg diet of DON, causes adverse effects on chicken growth performance [10]. Further, these subclinical doses of FB1 and DON in feed predispose broiler chickens to various diseases, including coccidiosis [11], necrotic enteritis [6], and salmonellosis [12], by compromising immune function and damaging gut integrity.

Currently, there are no biomarkers to detect subclinical mycotoxicosis in poultry, and identifying subclinical mycotoxicosis in poultry based on production performances is not practical. Therefore, it is important to identify biomarkers for FB1 and DON toxicity that can accurately pinpoint real-time mycotoxin exposure and toxicity to facilitate early intervention. Mycotoxins are metabolized to various degradation derivatives in the digestive tract, intestinal mucosa, liver, and kidneys [13]. Quantifying mycotoxins and their metabolites in various biological matrices, such as excreta [14], plasma or serum [14,15], muscle, liver, kidney [16,17,18], and feed [19,20], are currently available biomarkers for detecting mycotoxicosis. However, due to the lack of commercial standards, these methods have challenges in detecting phase I and II mycotoxin metabolites [21]. The mere quantification of FBs in feed or the liver cannot predict mycotoxicosis because chickens often do not exhibit clinical signs of toxicity. FB1 exposure can be identified by analyzing the altered ratio of the sphingolipids, particularly sphinganine (Sa)/sphingosine (So), in the serum and tissue [22]. In chickens, exposure to 20 mg/kg of FB1 for nine days caused detectable changes in the Sa/So ratio in muscle and liver tissues [23,24]. However, these doses did not reflect any detectable changes in the plasma Sa/So ratio [24]. In addition, feed processing methods, such as heat processing, increase protein-bound fumonisins. Although the protein-bound form is not toxic, free mycotoxins could be released into the digestive tract and cause toxicity [25,26]. Another concern is that although FBs were identified in feed, liver, and muscle, the birds exhibited no clinical signs of toxicity [27,28,29,30], thereby questioning the validity of utilizing serum or liver FBs content as a biomarker of FBs contamination in poultry. In poultry species, DON and its bio-transformed product, de epoxy-deoxynivalenol (DOM-1), are not considered ideal biomarkers because their levels are below the detection limits in chicken blood and excreta [31,32,33,34]. Further, LC-MS/MS and LC-HRMS analysis involve complex instrumentation, making it difficult to quantify the DON and other bio-transformed products, such as DON-3-sulfate and DON-15-sulfate [33]. This complexity may lead to an underestimation of mycotoxin levels in foodstuffs and an associated increase in health risks to consumers.

In poultry, the absorption of mycotoxins like FBs and DON varies from 1% to 6% and is associated with decreased gut integrity [35]. Additionally, FBs and DON undergo entero-hepatic circulation, resulting in reabsorption and prolonged retention in the gut. This limited absorption of FBs and DON [35,36,37] indicates that a substantial portion of non-absorbed toxin remains within the lumen of the gastrointestinal tract (GIT) and targets the gut epithelial cells. Since mycotoxicity leads to gut damage, assessing the intestinal permeability using FITC-d (4 kD) [38] can be used as an indicator for detecting both acute and chronic mycotoxicity. Among other modes of action, both FBs and DON directly inhibit protein synthesis, leading to decreased serum immunoglobulin levels and serum protein levels, as well as increased apoptosis in the liver. Hence, gut inflammatory markers and apoptosis-related markers could serve as potential biomarkers to identify FBs and DON exposure.

In recent years, microRNAs (miRNAs) have emerged as a promising class of biomarkers for monitoring toxicity and diseases in farm animals [39]. These biomarkers can show alterations in their expression before pathophysiological changes occur [40,41]. They are very stable in biofluids [42,43], and expression is often tied to inflammatory pathways [44]; thus, miRNAs can be desirable molecular biomarkers in response to acute environmental cues [45]. Studies have identified potential miRNA biomarkers for poultry diseases like Marek’s disease [46], avian leukosis [47], infectious bursal disease [48], avian influenza [49], and chicken necrotic enteritis [50]. Studies in chickens have also examined the role of miRNA in identifying the toxicity mechanisms of aflatoxin B1 in Roman laying hens [51] and ochratoxin A in broiler chickens [52]. However, to the best of our knowledge, there has been no research on utilizing miRNA as biomarkers of FB1, DON, and ZEA toxicity in broiler chickens. Identifying miRNA biomarkers of mycotoxicosis is underexploited and has great potential to allow non-destructive, real-time tracking of mycotoxin exposure in poultry. Thus, the objective of this study was to quantify the effects of different combinations of subclinical doses of FB1, DON, and ZEA on production performance, blood biochemistry, gut permeability on day 14 and day 21, and miRNA expression on day 14 to identify potential biomarkers for early mycotoxin exposure.

## 2. Results

### 2.1. Effect of Combined Doses of Mycotoxin on Production Performance

No significant difference (*p* > 0.05) was observed in BWG and mortality-adjusted feed conversion ratio (FCR) among treatment groups on day 7 and day 14 (Table 1). However, a significant difference (*p* < 0.05) was observed in BWG among treatment groups on day 21. Birds in the T2 groups, exposed to the lowest concentration of added mycotoxins in this experiment, had significantly lower BWG than birds in the control group. Birds in the T2 groups had 13% significantly lower 0–21 d BWG than birds in the control group. Birds exposed to higher concentrations of feed mycotoxins than in the T2 group had no further significant decrease in BWG compared to birds in the T2 group. Birds in the T2 to T6 groups had numerically decreased 0–21 d FCR compared to birds in the control group.

### 2.2. Effect of Combined Doses of Mycotoxin on FITC-d Permeability

On day 14, there were significant differences (*p* < 0.05) across treatment groups on serum FITC-d concentrations when compared to the control group (Figure 1). Birds in the control group had the lowest serum FITC-d concentrations, and birds in the T4, T5, and T6 groups had the highest serum FITC-d concentrations.

On day 21, there were significant differences (*p* < 0.05) in serum FITC-d concentrations across the treatment groups compared to the control group (Figure 1). Birds in the control group had the lowest serum FITC-d concentrations, and birds in the T6 group had the highest serum FITC-d concentrations. Birds in the T2 groups showed a significant one-fold increase in the serum FITC-d concentrations compared to the control group (*p* < 0.05). Birds in T3, T4, and T5 showed a further increase in the serum FITC-d concentrations compared to birds in the control group (*p* < 0.05). Birds exposed to increasing concentrations of feed mycotoxins showed a dose-dependent increase in FITC-d concentration, with higher mycotoxin concentrations leading to a greater decrease in gut integrity.

### 2.3. Effect of Combined Doses of Mycotoxin on Liver Apoptosis

On day 14, early, late, and total apoptotic cell percentages in the liver were significantly different (*p* < 0.05) between treatment groups (Figure 2). Birds in the T2 groups had 6.8-, 9.9-, and 8.5-fold significantly higher early, late, and total apoptotic cell percentages than the control group, respectively. Birds in T4, T5, and T6, which were exposed to higher concentrations of feed mycotoxins than birds in the T2 group, had no further increase in total apoptotic cell percentages in the liver.

### 2.4. Effect of Combined Doses of Mycotoxin on Blood Biochemistry

On day 14, serum glucose, phosphorus, and uric acid concentrations had no significant differences (*p* > 0.05) across the treatment groups compared to the control, whereas significant differences were observed across the treatment groups for serum AST and CK concentrations (*p* < 0.05) (Table 2). Birds in the T4 and T6 groups had significantly higher AST concentrations when compared to the control group (*p* < 0.05). Birds in T2, T4, T5, and T6 had significantly higher CK concentrations when compared to the control (*p* < 0.05) (Figure 3 and Figure 4).

On day 21, significant differences were observed in serum phosphorus, total protein, glucose, and potassium levels (*p* < 0.05) across the treatment groups, while calcium and uric acid levels showed no significant differences (*p* > 0.05) compared to the control group (Table 2). Birds in the T2 group had 57% significantly lower phosphorus, 40% significantly higher total protein, 8% significantly lower glucose, and 50% significantly lower potassium serum concentrations than birds in the control group. Birds exposed to higher concentrations of feed mycotoxins than the T2 group had no further decrease in serum phosphorus, glucose, and potassium serum concentrations compared to the T2 group. Birds in the T6 group had a further increase in serum total protein compared to the T2 group.

On day 21, serum AST and CK concentrations were significantly different (*p* < 0.05) across treatment groups when compared to the control (Figure 3 and Figure 4). Birds in the T2 group had significantly higher serum AST, and birds in the T4 group had significantly higher serum CK concentrations compared to the control group. Birds exposed to the lowest concentrations of feed mycotoxins also had elevated serum AST and CK concentrations.

### 2.5. Effect of Combined Doses of Mycotoxin on Liver miRNA Expression

On day 14, liver gga-let-7a-5p, gga-miR-19b-3p, gga-miR-9-5p, gga-miR-217-5p, gga-miR-133a-3p, gga-miR-215-5p, and gga-miR-29c-3p miRNA concentrations were significantly (*p* < 0.05) different between the treatment groups (Table 3). Birds in T6, exposed to the highest concentration of mycotoxins in this experiment, had significant upregulation in liver gga-let-7a-5p (14.2-fold), gga-miR-19b-3p (2.72-fold), gga-miR-9-5p (7.1-fold), gga-miR-217-5p (16.9-fold), gga-miR-133a-3p (7.4-fold), and gga-miR-215-5p (7.0-fold) compared to the control group. Birds in T3 had significant increases in gga-miR-9-5p (3.8-fold) and gga-miR-217-5p (3.2-fold) compared to the control group. The miRNAs gga-miR-155, gga-miR-375, and gga-miR-29c-3p showed no statistically significant changes in their expression across treatments (*p* > 0.05).

### 2.6. Effect of Combined Doses of Mycotoxin on Liver Histopathology

In the liver histopathological scoring on day 21, T3 had mild lesions of hemorrhage, hepatocyte degeneration, vacuole, presence of inflammatory cells, fibrosis, and bile duct proliferation; T2 and T5 had moderate lesions; T4 and T6, the highest DON and FB1 containing groups, had the most severe lesions compared to the control T1 as shown in Table 4. Haematoxylin and eosin-stained liver images are shown in Figure 5.

## 3. Discussion

This study identified possible biomarkers for exposure to subclinical doses of combined FB1, DON, and ZEA. According to the BIOMIN Mycotoxin Survey 2023, approximately 92% of corn and corn by-products [8] were contaminated with more than one mycotoxin, contributing to a USD 900 million annual economic loss due to the contamination of FBs, DON, ZEA, and aflatoxin B1. Hence, this study aimed to identify biomarkers for combined exposure to subclinical mycotoxins and exposure to major mycotoxins, including FB1, DON, and ZEA, that could be detected in commercial settings as early as 14 d after exposure. In the current study, birds exposed to subclinical doses of 2.0 mg FB1 + 2.5 mg DON + 0.9 mg ZEA/kg diet had decreased BWG at d 21, increased gut FITC-d permeability, higher early, late, and total apoptotic cell percentages, and lower phosphorus, glucose, and potassium, and higher total protein in the serum by day 21. On d 14, broiler chickens exposed to subclinical concentrations of 21.0 mg FB1 + 3.0 mg DON + 1.0 mg ZEN/kg diet had higher expressions of miRNA gga-let-7a-5p, gga-miR-19b-3p, gga-miR-9-5p, gga-miR-217-5p, gga-miR-133a-3p, and gga-miR-215-5p in the liver.

The selected mycotoxin concentrations aimed to simulate real-world contamination levels encountered in commercial poultry production, ranging from low risk to high levels of subclinical exposures without exceeding FDA guidance limits. However, the variability in mycotoxin concentrations across treatment groups reflects real-world challenges, as achieving consistent toxin ratios, especially with DON to ZEA culture materials, is quite challenging. Despite thorough mixing to minimize variability, “hot spots” mimic practical feed contamination scenarios. While precise dose–response effects are influenced by environmental factors such as temperature and humidity, observed trends in this present study, particularly at higher toxin concentrations, suggest a general dose-dependent relationship. This approach highlights the complexities of co-contamination and provides insights into subclinical mycotoxin exposure risks in poultry.

In the current study, FB1, DON, and ZEA concentrations were below the FDA guidance levels. As mycotoxins are ubiquitous in nature, even though clean corn was used to produce the basal diet, natural contamination in the finished diet reflects the field conditions. Hence, in this current study, the starter basal diet in the control group was naturally contaminated with 0.4 mg/kg FB1 and 0.6 mg/kg DON [10]. The mycotoxin levels in the basal diet (FB1; 0.4 mg/kg and DON; 0.6 mg/kg) were much lower than the regulatory guidance levels (e.g., FDA limits: 50 mg/kg FB1 and 5 mg/kg DON for broiler finished feed). Considering all experimental groups, including the control, were fed the same baseline diet, the background contamination had an identical impact on each group. This guarantees that the variation observed between treatment groups can still be caused by the additional mycotoxin exposure introduced in the treatments. Subclinical concentrations of mycotoxins had no impact on production performances until d 14. However, on d 21, the BWG was decreased by 13%, even at the lowest subclinical concentration of 2 mg FB1 + 2.5 mg DON + 0.9 mg ZEA/kg diet. When the mycotoxin concentration was increased to 21 mg FB1 + 3 mg DON + 1 mg ZEA/kg diet, it further decreased the BWG up to 17%. These findings are consistent with previous research by Kubena et al. (1997) [53], who reported a 19% reduction in BWG on d 21 when chickens were exposed to 300 mg FB1+ 15 mg DON per kg diet. Similar results were observed in broiler chickens exposed to 20 mg FUM + 1.5 mg DON/kg diet and 20 mg FUM + 5.0 DON/kg diet for 21 d, resulting in a 6% decrease in BWG [54]. The presence of multiple mycotoxins in poultry diets, even at low doses, had a negative effect on chicken production performance [10,54], and earlier studies reported the negative effects of DON on production performance at doses only above 5 mg/kg diet [55]. In this present study, even in groups with a diet containing 2 mg FB1 + 2.5 mg DON + 0.9 mg ZEA/kg diet, there is a reduction in BWG. This significant decrease in BWG among the mycotoxin treatment groups suggests that because of the lowest bioavailability of FBs and DON, even the low levels of combined toxins in the feed likely contribute to gut damage. This intestinal damage is most likely linked to reduced nutrient absorption and increased maintenance demands on the gut, leading to higher metabolic costs for tissue repair and immune system activation, ultimately resulting in decreased BWG in chickens.

In the present study, even the lowest concentration of mycotoxins (2 mg FB1 + 2.5 mg DON + 0.9 mg ZEA/kg diet) increased gut permeability, with higher doses further increasing gut permeability both on d 14 and d 21. These results suggest that the presence of DON as low as 2.5 mg/kg feed can compromise gut integrity irrespective of FB1 and ZEA concentration. A similar finding was observed when broiler chickens were exposed to 3 mg FB1 + 4 mg DON per kg diet [10] and/or 5 mg/kg DON alone, resulting in compromised intestinal integrity [56]. The absorption of FB1 and DON in poultry is low, typically ranging from 1% to 6% [35], meaning that a substantial portion of feed mycotoxin remains in the gastrointestinal tract [36,37]. In addition, FB1 and DON undergo entero-hepatic circulation, resulting in prolonged exposure of gut epithelial cells to these mycotoxins [38]. Mycotoxins induce oxidative stress [57] and inflammatory responses [58], which further compromise gut health. Gut microbiota can biotransform mycotoxins, such as DON, FB1, and ZEA, into secondary metabolites, which may have less, similar, or even greater toxicity. These metabolites could potentially act on the gut lining, leading to inflammation and increased permeability [59]. In this study, a general tendency for a dose-dependent increase in serum FITC-d was observed, which indicates that the FITC-d assay can be a potential biomarker for evaluating d 14 and d 21 when combined mycotoxin exposure concentrations, even at 2.0 FB1 + 2.5 DON + 0.9 ZEA mg/kg diet.

Subclinical exposure to multiple mycotoxins causes damage to internal organs without causing overt clinical signs. In this study, exposure to even low concentrations of 2 mg FB1 + 2.5 mg DON + 0.9 mg ZEA/kg diet was sufficient to increase AST concentrations in all experimental groups on d 21, whereas, on d 14, concentrations with high FB1 of 21 mg/kg (T6) and high DON of 3.5 mg/kg (T4) showed a significant increase in serum AST. Elevated AST concentrations correlated with mycotoxin-induced histological changes in liver tissues that included necrosis, apoptosis, and inflammatory infiltrates. Hepatocellular injury releases AST into the circulation [60], and thus, increased AST concentrations reflect significant hepatocellular damage and compromised liver function [61,62]. Mycotoxins can interfere with various hepatic metabolic pathways, including gluconeogenesis [63]. In the present study, histological changes in liver tissues, including necrosis, apoptosis, and inflammatory infiltrates in all the treatment groups, correlated with elevated AST levels across all treatment groups.

CK is crucial for maintaining cellular energy homeostasis by catalyzing the transfer of a phosphoryl group from ATP to creatine to produce phosphocreatine (PCr), which acts as a buffer and transporter of energy [64]. PCr shuttles energy to ATP-dependent enzymes like Na^+^, K^+^-ATPase, and H^+^-ATPase, which are essential for cellular ionic balance and pH regulation [65]. Studies suggest that mycotoxins impair the CK/PCr system, reducing ATP availability and thus compromising ATPase activities, affecting Na^+^, K^+^, and Ca^2+^ balance, and disrupting ATP production and energy homeostasis [66]. In the current study, an increased concentration of CK was observed in birds exposed to even the lowest doses of mycotoxins (2 mg FB1 + 2.5 mg DON + 0.9 mg ZEA/kg feed) as early as day 14 and continuing through day 21. This clearly indicates that combined mycotoxins, even at subclinical doses, affect CK activity. Thus, elevated serum CK indicates extracellular leakage of the enzymes into the bloodstream due to tissue injury and intracellular inhibition of CK activity, reducing the availability of ATP and compromising Na^+^, K^+^-ATPase, and H^+^-ATPase activity. A previous meta-analysis study identified that exposure to 0.95 mg/kg for aflatoxins or 4.29 mg/kg DON, 2.87 mg/kg T2 toxin, 0.78 mg/kg ochratoxins, or 5.05 mg/kg ZEA or 112.80 mg/kg FUM can disrupt metabolic pathways and cause hepatic and kidney damage [67]. Our study findings suggest that elevated serum AST and CK may be biomarkers of subclinical mycotoxicity. However, more investigation is required to understand the ontogeny of biomarker changes and whether these changes are specific to an individual or a combination of mycotoxins.

In the present study, serum phosphorus (P) levels were significantly decreased at low concentrations (2 mg FB1 + 2.5 mg DON + 0.9 mg ZEA/kg diet) on d 21. This reduction is most likely due to a cytoprotective mechanism where serum P is used to produce phosphorylated sphingosine, which serves as a defense against the toxic effect of FB1 [68]. Additionally, the ingestion of fusarium mycotoxins impairs kidney function, which affects P homeostasis and causes increased excretion of P in urine, causing a decrease in serum P concentration [69]. P availability is critical for physiological functions such as bone development, and the reduction in P availability across all treatment groups may contribute to the decreased growth performance observed in those birds [70,71]. Blood glucose concentrations were also significantly lower in all treatment groups compared to control groups. The T4 treatment groups, which had a DON concentration of 3.5 mg/kg, showed a greater reduction in glucose levels. DON is known to decrease gut integrity and downregulate the expression of sodium-dependent glucose transporters (SGLT-1, GLUT-2) in the intestine [72,73], leading to a decrease in blood glucose concentrations [74]. A similar result was observed when the pigs exposed to a combined dose of DON 1.0 mg + ZEA 1.04 mg/kg diet decreased the sucrase, maltase, and lactase enzyme activities in the intestine [75]. This suggests that when the mycotoxin concentrations are even at 2 mg FB1 + 2.5 mg DON + 0.9 mg ZEA/kg diet, they are sufficient to decrease the blood glucose levels. There were no significant differences in d 14 serum phosphorus and glucose concentrations, which shows that these cannot be potential biomarkers for detecting day 14 exposure.

In this study, 21-day exposure to FB1 + DON + ZEA had a significant impact on potassium homeostasis across all treatment groups, but there was no significant effect on day 14 potassium concentrations, which shows that serum potassium concentration can be a potential biomarker for the lowest mycotoxin concentrations of 2 mg FB1 + 2.5 mg DON + 0.9 mg ZEA/kg diet on day 21. Similar results were observed when broiler chickens were exposed to 1.7 mg DON + 0.2 mg ZEA/kg diet for 35 days [76]. Serum potassium level is considered an indicator of protein metabolism, and decreased levels are associated with decreased catabolism and nephrotoxicity. Potassium is primarily reabsorbed in the kidneys, particularly in the proximal tubules, and is secreted in the distal tubules [77]. The decreased serum potassium level observed in this study is most likely linked to impaired renal function due to multiple mycotoxin exposure. In addition, potassium is an indicator of overall metabolic health, and the decreased serum levels could indicate disruptions in protein metabolism and overall catabolism, leading to significant biochemical changes in blood parameters, including alterations in electrolyte balance [61,62].

In this study, we also aimed to assess whether combined exposure to mycotoxins altered miRNA expression before the presence of observable damage in the gut and liver. We tested miRNA expression on d 14, prior to any significant histological changes or functional impairments in these organs. The results showed that even at this early time point, exposure to 21 mg FB1 + 3 mg DON + 1 mg ZEA/kg diet caused a significant upregulation of liver miRNAs, including gga-let-7a-5p, gga-miR-19b-3p, gga-miR-9-5p, gga-miR-217-5p, gga-miR-133a-3p, and gga-miR-215-5p. These changes in miRNA expression suggest that alterations in gene regulation can occur before any overt damage to the gut or liver, highlighting the potential of miRNAs as early biomarkers for detecting mycotoxin exposure and its subclinical effects. Additionally, studies on these miRNAs in poultry and other species indicated that gga-miR-155 and gga-miR-9-5p play an important role during inflammation [78,79,80]. miRNAs such as gga-miR-29c-3p and gga-miR-217-5p are involved in apoptosis [81,82,83], and gga-miR-375 is related to oxidative stress [84]. Further, gga-miR-19b-3p and gga-miR-133a-3p are involved in lipid metabolism and growth [85,86,87], and gga-miR-215-5p, and gga-let-7a-5p have roles in the cell differentiation process [88,89,90]. The expression of the gga-miR-155, gga-miR-375, and gga-miR-29c-3p miRNAs was not affected by the mycotoxin exposure levels analyzed in this study. Increased expression of gga-let-7a-5p miRNA indicates that mycotoxins cause stress to hepatocytes and promote liver apoptosis. Similarly, upregulation of gga-miR-217-5p expression suggests the presence of oxidative stress since gga-miR-217-5p stimulates apoptosis-related pathways in order to remove any damaged cells and maintain cell homeostasis in the liver [83]. The overexpression of gga-miR-19b-3p suggests that mycotoxins are causing inflammation in the hepatocytes by activating the NF-κB signaling, which in turn increases the production of inflammatory cytokines [91]. miRNA gga-miR-215-5p controls cell cycle and proliferation [92], and overexpression of gga-miR-215-5p can be a host compensatory response to mitigate cell damage caused by mycotoxins. Studies have also shown that, both in vitro and in vivo, mycotoxicosis can modify the expression of specific miRNAs [93]. ZEA exposure in pigs modifies ssc-miR-455-5p, 493-3p, 135a-5p, 432-5p, 542-3p, and 493-3p, and aflatoxin B1 exposure modifies gga-miR-301a-3p and gga-miR-301b-3p miRNA in poultry [51]. These modifications suggest that miRNAs are potential biomarkers for detecting mycotoxin exposure. These above studies suggest that the expression pattern of miRNAs could serve as a potential marker to track subclinical mycotoxicity and disease severity in chickens.

Out of nine candidate miRNAs, gga-let-7a-5p, gga-miR-19b-3p, gga-miR-9-5p, gga-miR-217-5p, gga-miR-133a-3p, and gga-miR-215-5p can be used as biomarkers for detecting subclinical concentrations of combined mycotoxin exposure, particularly at the 21 mg FB1 + 3 mg DON + 1 mg ZEA/kg diet (T6). These combined doses of mycotoxins significantly upregulate miRNAs, suggesting that these miRNAs are particularly reliable for detecting mycotoxin exposure. While serum FITC-d, AST, and CK levels provide valuable information about mycotoxin-induced damage even at subclinical doses, changes in miRNA expression detected at day 14 suggest that miRNAs are more sensitive and timelier tools for identifying early disruptions in gene regulation. Further histological changes, such as necrosis, apoptosis, and gut integrity, were observed at day 21 and were preceded by alterations in miRNA expression at day 14. These results suggest that miRNA could serve as an early biomarker for subclinical mycotoxicity and a useful molecular tool for detecting mycotoxin exposure before severe pathological damage. This ability to detect changes in miRNA expression as early as 14 days after exposure would help farmers take preventive measures and reduce economic losses.

In commercial settings, chickens are rarely exposed to individual mycotoxins. Therefore, the findings from this study provide insights into the combined effects of mycotoxins under practical conditions, even if precise interactions (synergistic, additive, or antagonistic) could not be separately evaluated. While single-toxin treatment groups could have allowed us to analyze specific interactions, outcomes vary depending on several factors, including toxin type, dosage, and duration of exposure. For example, some studies have demonstrated that the co-occurrence of DON and ZEA has a synergistic effect on intestinal integrity [75,94,95], while other combinations may show antagonistic effects [75,96]. Hence, results of the current study are especially relevant to poultry, where chickens are exposed to several mycotoxins.

In summary, subclinical exposure to combined FB1 + DON + ZEA (<5 mg/kg DON and 50 mg/kg FB1) increased intestinal permeability by 1-fold, liver total apoptotic cells by 8-fold, decreased serum P by 57%, increased serum total protein by 40%, decreased serum glucose by 8%, and decreased serum potassium concentrations by 50% on day 21. Serum FITC-d could serve as a potential biomarker for detecting mycotoxin doses of 9 FB1 +3.5 DON +0.7 ZEA and above for 14 d exposure. However, miRNAs gga-let-7a-5p, gga-miR-19b-3p, gga-miR-9-5p, gga-miR-217-5p, and gga-miR-215-5p by several folds, these miRNAs could serve as potential biomarkers at doses of 21 mg FB1 + 3 mg DON + 1 mg ZEA/kg diet for 14 d exposure. However, further research is needed to validate the expression patterns of these liver miRNAs with those in serum and jejunum. This will not only help us to understand the molecular mechanisms involved in the mycotoxin-induced apoptosis but also allow early diagnosis of mycotoxin exposure, thereby facilitating timely intervention strategies.

## 4. Conclusions

This study emphasized that a chicken diet containing multiple mycotoxins below FDA recommendations led to a range of physiological and biochemical changes in broiler chickens, including increased intestinal permeability, liver apoptosis, and altered serum parameters. The upregulation of miRNAs such as gga-let-7a-5p, gga-miR-19b-3p, gga-miR-9-5p, gga-miR-217-5p, and gga-miR-215-5p on d 14 suggests that these can be potential biomarkers for early detection of subclinical mycotoxicosis. Additionally, the significant increases in serum FITC-d, AST, and CK concentrations on both days 14 and 21 further support their role as biomarkers for the early detection of mycotoxin exposure, and future studies should focus on validating these miRNAs and serum enzymes in other tissue samples and serum or plasma, making it an economical approach to biomonitor mycotoxins in real time.

## 5. Materials and Methods

### 5.1. Birds, Diet, and Animal Health

A total of 720 one-day-old male Ross × Ross 708 strain broiler chicks vaccinated at day 0 against Eimeria using CocciVAC (Aviagen, Blairsville, GA, USA) were used for conducting a 21-day feeding trial. All animal protocols were approved by the Institutional Animal Care and Use Committee at the Southern Poultry Research Group, Athens, GA. Day-old broiler chicks were raised in 1.5 m × 1.5 m floor pens (stocking density of 20 birds/m^2^) on new litter. In accordance with North American industry standards, the floor pens were equipped with thermostatically controlled heaters and nipple-style waterers. The birds were fed with corn- and soybean-based basal diet. (Table 5). FB1, DON, and ZEA were produced on rice culture using *F. verticillioides* (FRC M3125) and *F. graminearum* (PH-1), as described earlier [10]. The *F. graminearum* (PH-1) produced ZEA in addition to DON, so ZEA was present in the experimental treatment groups at concentrations ranging from 0.1 to 1 mg/kg of diet. The homogenized rice cultures were mixed with a basal diet to make experimental diets. The chicks were weighed individually and randomly assigned to one of the six treatment groups with increasing concentrations of mycotoxins. The final finished diets were analyzed by LC-MS-MS to determine FB1, DON, and ZEA (Romer Labs, Union, MO, USA; Table 6). Other mycotoxins and their metabolites were not detected in the samples, as their concentrations were below the detection limits of the analytical methods used.

### 5.2. Treatment Groups

The control group (T1) had the lowest concentration of mycotoxins, and T6 had the highest concentration of mycotoxins. The concentration of mycotoxins in the feed was intended to increase gradually from lowest to highest from the treatment T2 to T6, but the feed analysis showed unexpected variation with different concentrations in the final analyzed diet. This variation may be due to the hotspots in the feed, where the mycotoxins are not evenly distributed, resulting in high or low concentrations during the sampling of the feed. The experimental treatment groups were T1: Control group (basal diet containing 0.6 mg FB1 + 0.4 mg DON + 0.0 mg ZEA/kg diet), T2: 2.0 mg FB1 + 2.5 mg DON + 0.9 mg ZEA/kg diet, T3: 5.0 mg FB1 + 0.4 mg DON + 0.1 mg ZEA/kg diet, T4: 9.0 mg FB1 + 3.5 mg DON + 0.7 mg ZEA/kg diet, T5: 17.0 mg FB1 + 1.0 mg DON + 0.2 mg ZEA/kg diet and T6: 21.0 mg FB1+ 3.0 mg DON + 1.0 mg ZEA/kg diet. Each treatment was replicated in six pens with 20 birds per pen in a completely randomized design. Chicks had ad libitum access to feed and water throughout the experimental period.

### 5.3. Production Performance and Sampling

On days 0, 7, 14, and 21, body weight and feed intake were measured. The average feed intake and BWG were corrected for mortality and for calculating the FCR for each pen. On days 14 and 21, one bird per pen (*n* = 6) was selected randomly from each pen to maintain sample representativeness and was euthanized by cervical dislocation. Sampling on day 14 was performed to analyze early miRNA responses, while day 21 was chosen for systemic markers, as physiological changes generally manifest around this time. A small section of the liver was collected in RNAlater® (Sigma Aldrich, St. Louis, MO, USA) for miRNA analysis, histology, and blood samples for blood chemistry analysis.

### 5.4. Intestinal Permeability Assay and Sampling

On days 14 and 21, intestinal permeability was measured using FITC-d (4000 Da; Sigma-Aldrich, St. Louis, MO, USA), as described earlier [10]. One bird from each pen (*n* = 6) was orally gavaged with 2.2 mg/mL FITC-d (MW 4000; Sigma-Aldrich, ON, Canada). Two hours later, the birds were sacrificed, and blood was collected from the heart and stored in opaque tubes. Blood samples were centrifuged at 450× *g* for 10 min to separate the serum. Then, 100 mL of blood serum was added in duplicates for each sample on a black opaque flat-bottomed 96-well plate. A standard curve was generated with different concentrations of known FITC-d concentration on the same plate as the samples using Gen5 Microplate Reader and Imager software (version 3.10). The FITC-d in the serum samples and standards was measured at an excitation wavelength of 485 nm and an emission wavelength of 528 nm using a microplate reader (Synergy HT, multi-mode microplate reader, BioTek Instruments, Inc., Vermont, VT, USA).

### 5.5. Liver Apoptosis Assay

On day 14, hepatocytes were isolated by density gradient centrifugation using Ficoll-Paque media (Sigma-Aldrich, St. Louis, MO, USA). 100 μL of 1 × 10^6^ hepatocytes from each sample (*n* = 6) were incubated with 100 μL of Guava Nexin reagent (Cytek, Fremont, CA, USA) and incubated for 20 min according to the manufacturer’s instructions. The Guava Nexin Reagent contains two dyes, Annexin-V-PE and 7-AAD. The percentage of apoptotic cells from each sample was analyzed in a flow cytometer (Guava EasyCyte, Millipore, MA, USA), and the percentage of apoptotic cells was calculated using Guava Nexin software (version 4.5). The results were expressed as the percentage of early apoptotic cells (AnnexinV^+^,7-AAD^-^), late apoptotic cells (AnnexinV^+^,7-AAD^+^), and total apoptotic cells (Early + late apoptotic cells).

### 5.6. Blood Biochemical Analysis

On days 14 and 21, one bird per pen (*n* = 6) was randomly selected, and blood samples were collected from the brachial vein without an anticoagulant. The serum was analyzed for phosphorus, total protein (TP), glucose (GLU), potassium (K^+^), calcium (Ca^2+^), uric acid (UA), creatine kinase (CK), and aspartate aminotransferase (AST) using the Vetscan^®^ VS2 Chemistry Analyzer (Abaxis, Inc., Union City, CA, USA).

### 5.7. MicroRNA Expression Analysis

On day 14, liver gga-miR-29c-3p, gga-miR-217-5p, gga-miR-375, gga-miR-215-5p, gga-miR-155, gga-miR-9-5p, gga-miR-19b-3p, gga-miR-133a-3p, and gga-let-7a-5p miRNA concentrations were analyzed by real-time qPCR. Total RNA, containing miRNA, was isolated from liver samples using the miRNeasy kit (Qiagen, Germantown, MD, USA) following the manufacturer’s instructions. The quality of the RNA was evaluated by using a Bioanalyzer, 2100 Expert Software (B. 02. 08), version 2.6 (Agilent Technologies, Santa Clara, CA, USA) using an RNA 6000 Nano Kit (Agilent Technologies, Santa Clara, CA, USA). The RNA integrity number (RIN) for each sample was above 7.5 (Figure 6). One sample in the T3 group with a RIN below 7.5 was excluded from cDNA synthesis. cDNA was synthesized with 2 μg of RNA using a miRNA 1st strand cDNA synthesis kit (Agilent Technologies, Santa Clara, CA, USA). Real-time PCR was carried out using the miRNA qRT-PCR detection kit (Agilent Technologies, Santa Clara, CA, USA) with a universal reverse primer. A total of nine miRNA genes and one internal control gene, gga-U6 (used as a housekeeping gene for normalizing miRNA expression), were amplified using specific forward primers and a universal reverse primer (Table 7). Each well contained 12.5 µL of 2× miRNA QPCR master mix (Agilent Technologies, Santa Clara, CA, USA), 8.5 µL of nuclease-free water, 1 µL of cDNA (2 µg/µL), 1 µL of miRNA specific forward primer (3.125 µM), and 1 µL of universal reverse primer (3.125 µM). The settings for real-time PCR were 95 °C for 10 min (1 cycle), followed by 95 °C for 15 s and 60 °C for 45 s (45 cycles). The melting profile was determined by heating samples at 65 °C for 30 s and then increasing the temperature at a linear rate of 10 °C/s to 95 °C while continuously monitoring fluorescence (CFX Maestro software, version 2.3). The data were normalized with reference gene gga-U6. The fold change was calculated using the 2^−∆∆Ct^ method to determine the differential expression of miRNA genes. The control group was used as a reference group.

### 5.8. Histopathological Analysis

On day 21, liver samples from each treatment group were fixed in a 10% buffered formalin solution and processed using a tissue processor as described earlier [6]. The samples were processed in a graded series of alcohols (30 min each in 50%, 70%, 95% ethanol, and 100% ethanol with one change at 30 min) at room temperature. Samples were cleared using Pro-par (Anatech, Battle Creek, MI, USA) for 45 min with two changes at 30 min, followed by infiltration with paraffin overnight at 60 °C with one change at 30 min using a tissue processor (Sakura Finetek USA, Inc., Torrance, CA, USA). Then, the samples were embedded in paraffin blocks. These blocks were cut into 5-μm cross-sections and were mounted on superfrost slides (Thermo Fisher Scientific, Waltham, MA, USA), then stained with hematoxylin and eosin. A total of 10 fields/treatments were randomly observed under an Olympus BX60 brightfield microscope (Olympus Corp., Tokyo, Japan), and changes in the liver histopathological parameters, including fibrosis in the portal area, bile duct proliferation, hepatocyte degeneration, and inflammatory cellular infiltration, were observed under 20× magnification and scored as no changes (−), mild (+), moderate (++), and severe (+++), respectively, as shown in Table 4 [100].

### 5.9. Statistical Analysis

A one-way ANOVA (JMP Pro software version 16.2.0, SAS Institute Inc., Cary, NC, USA) was conducted to analyze the effects of different concentrations of combined doses of mycotoxins on the dependent variables, with the pen being considered the experimental unit. When the main effects were significant (*p* < 0.05), differences between means were analyzed by Tukey’s least-square means comparison. Values are reported as means + SEM.

## Figures and Tables

**Figure 1 toxins-17-00001-f001:**
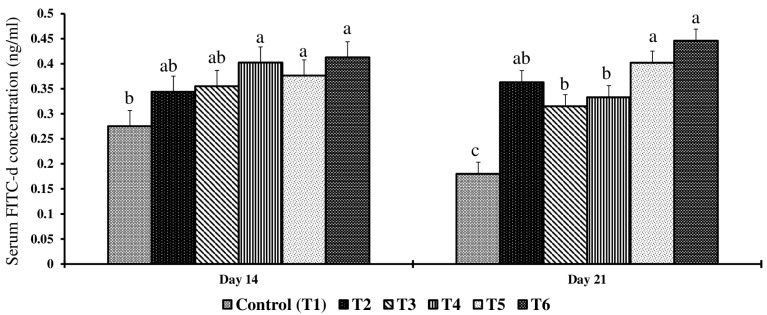
Effect of combined doses of mycotoxin on gut permeability on day 14 and day 21. Bars (+SEM) with no common superscript differ significantly (*p* < 0.05) (*n* = 6).

**Figure 2 toxins-17-00001-f002:**
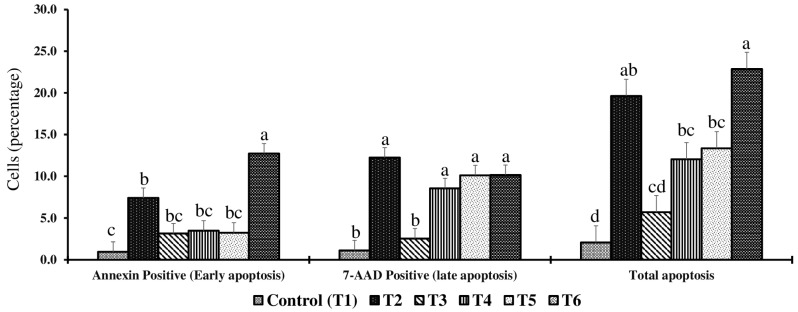
Effect of combined doses of mycotoxin on the percentage of early, late, and total apoptotic hepatocytes on day 14. Bars (+SEM) with no common superscript differ significantly (*p* < 0.05) (*n* = 6).

**Figure 3 toxins-17-00001-f003:**
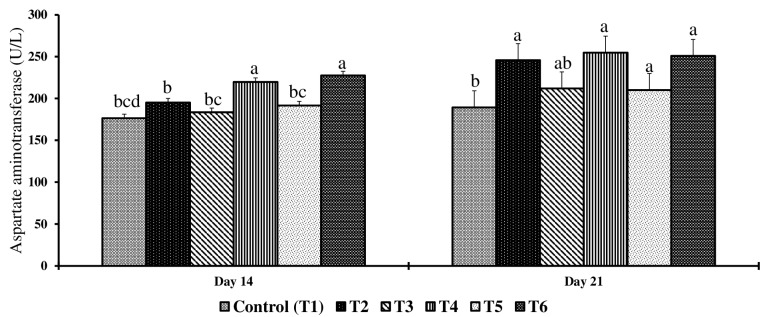
Effect of combined doses of mycotoxin on serum aspartate aminotransferase on day 14 and day 21. Bars (+SEM) with no common superscript differ significantly (*p* < 0.05) (*n* = 6).

**Figure 4 toxins-17-00001-f004:**
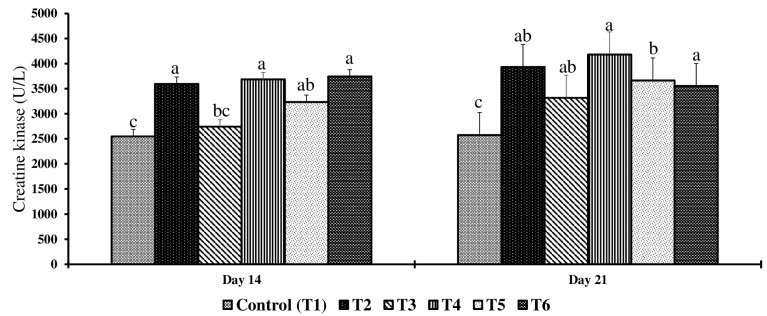
Effect of combined doses of mycotoxin on creatine kinase on day 14 and day 21. Bars (+SEM) with no common superscript differ significantly (*p* < 0.05) (*n* = 6).

**Figure 5 toxins-17-00001-f005:**
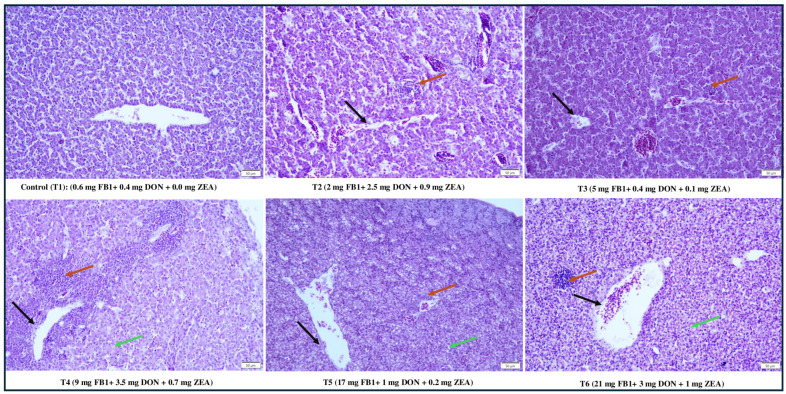
Effect of combined doses of mycotoxin on liver histopathology on day 21. Liver histology from a broiler chicken contaminated with FB1, DON, and ZEA for 3 weeks. Control (T1) has the normal histological appearance of the liver from a control broiler chicken (fed with a basal diet). Fibrosis and bile duct proliferation (red arrows) in portal areas and hepatocellular infiltration (black arrows) and vacuole (green arrows) were observed in T2, T3, T4, T5, and T6. Haematoxylin and eosin staining; Scale bar = 50 μm.

**Figure 6 toxins-17-00001-f006:**
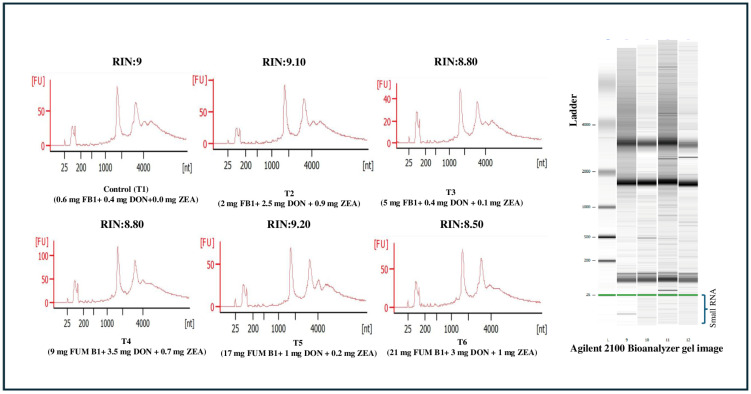
Total RNA integrity of treatment groups and their RIN. The RNA integrity number (RIN) was analyzed for each sample in the treatment using a Bioanalyzer 2100 instrument (Agilent Technologies, Santa Clara, CA, USA) using a small RNA chip RNA 6000 Nano Kit (Agilent Technologies, Santa Clara, CA, USA). The RIN values for each treatment group are control T1 (0.6 mg FB1 + 0.4 mg DON + 0.0 mg ZEA/kg diet): 9, T2 (2.0 mg FB1 + 2.5 mg DON + 0.9 mg ZEA/kg diet): 9.1, T3 (5.0 mg FB1 + 0.4 mg DON + 0.1 mg ZEA/kg diet): 8.8, T4 (9.0 mg FB1 + 3.5 mg DON + 0.7 mg ZEA/kg diet): 8.8, T5 (17.0 mg FB1 + 1.0 mg DON + 0.2 mg ZEA/kg diet): 9.2 and T6 (21.0 mg FB1+ 3.0 mg DON + 1.0 mg ZEA/kg diet): 8.5.

**Table 1 toxins-17-00001-t001:** Effect of combined doses of mycotoxin on body weight gain (BWG) and mortality-adjusted feed conversion ratio (FCR) on days 0, 7, 14, and 21. Mean ± SEM with no common superscript differs significantly (*p* < 0.05) (*n* = 6).

Day	Parameter	Control (T1)	T2	T3	T4	T5	T6	SEM	*p* Value
0–7 d	BWG (g)	56.30	55.40	47.90	55.90	54.20	49.30	3.20	0.42
	FCR	2.33	2.32	2.63	2.33	2.50	2.47	0.16	0.43
0–14 d	BWG (g)	213.30	202.80	219.90	187.00	211.90	195.90	12.60	0.37
	FCR	1.74	1.79	1.89	1.89	1.71	2.03	0.10	0.40
0–21 d	BWG (g)	461.20 ^a^	399.60 ^bc^	419.00 ^b^	404.60 ^bc^	408.10 ^bc^	383.10 ^c^	6.50	<0.05
	FCR	1.68	1.77	1.72	1.72	1.74	1.79	0.07	0.30

**Table 2 toxins-17-00001-t002:** Effect of combined doses of mycotoxin on serum biochemistry on day 14 and day 21. Mean ± SEM with no common superscript differs significantly (*p* < 0.05) (*n* = 6). Asterisk (*) indicates the result is outside of the reference interval of the VetScan^®^ VS2 Chemistry Analyzer.

Day	Parameter	Control (T1)	T2	T3	T4	T5	T6	SEM	*p* Value
D14	Phosphorus (mmol/L)	0.68	0.68	0.68	1.10	0.58	0.65	0.04	0.76
	Total Protein (g/dL)	<2.00 *	<2.00 *	<2.00 *	2.10	<2.00 *	2.60	-	-
	Glucose (mmol/L)	12.56	10.43	12.38	9.80	8.45	12.05	1.77	0.77
	Potassium (mmol/L)	1.6	2.3	1.8	2.1	1.5	2.3	0.9	0.48
	Calcium (mmol/L)	<1.0 *	<1.0 *	<1.0 *	<1.0 *	<1.0 *	<1.0 *	-	-
	Uric Acid (mmol/L)	0.22	0.24	0.20	0.27	0.23	0.27	0.03	0.34
D21	Phosphorus (mmol/L)	0.15 ^a^	0.07 ^c^	0.09 ^b^	0.08 ^b^	0.08 ^a^	0.06 ^a^	0.01	0.04
	Total Protein (g/dL)	2.00 ^c^	3.70 ^a^	3.00 ^b^	3.00 ^b^	2.50 ^c^	2.80 ^b^	0.06	0.03
	Glucose (mmol/L)	6.67 ^a^	6.19 ^b^	6.23 ^b^	5.70 ^c^	6.09 ^b^	6.09 ^b^	0.29	0.02
	Potassium (mmol/L)	7.80 ^a^	4.20 ^b^	4.00 ^b^	4.00 ^b^	3.90 ^b^	3.90 ^b^	0.05	0.04
	Calcium (mmol/L)	0.35	0.35	0.36	0.33	0.34	0.33	0.08	0.72
	Uric Acid (mmol/L)	0.19	0.19	0.17	0.20	0.19	0.18	0.08	0.65

**Table 3 toxins-17-00001-t003:** The effect of combined doses of mycotoxin on liver miRNA expression on day 14. The miRNA expression concentration was analyzed after correcting for the housekeeping gene gga-miR-U6 miRNA concentration and normalizing it to the miRNA content of the control group on day 14. All the mean values represent fold changes compared to the control group. Mean ± SEM with no common superscript differs significantly (*p* < 0.05) (*n* = 6).

Treatment	gga-let-7a-5p	gga-miR-19b-3p	gga-miR-155	gga-miR-9-5p	gga-miR-217-5p	gga-miR-133a-3p	gga-miR-375	gga-miR-215-5p	gga-miR-29c-3p
Control (T1)	1.00 ± 2.98^b^	1.00 ± 0.43^ab^	1.00 ± 1.58	1.00 ± 1.35^b^	1.00 ± 3.01^b^	1.00 ± 0.36^bc^	1.00 ± 0.29	1.00 ± 0.27^b^	1.00 ± 0.56
T2	0.29 ± 2.98^b^	0.14 ± 0.43^b^	0.45 ± 1.58	0.28 ± 1.35^b^	0.45 ± 3.01^b^	0.25 ± 0.36^c^	0.01 ± 0.29	0.33 ± 0.27^b^	0.08 ± 0.56
T3	0.12 ± 3.26^b^	0.17 ± 0.48^b^	0.16 ± 1.73	0.61 ± 1.47^b^	0.24 ± 3.30^b^	0.31 ± 0.40^c^	0.02 ± 0.31	0.07 ± 0.29^b^	0.06 ± 0.61
T4	1.38 ± 2.98^ab^	1.41 ± 0.43^ab^	3.80 ± 1.58	0.18 ± 1.35^b^	1.17 ± 3.01^b^	2.4 ± 0.36^b^	0.47 ± 0.29	0.54 ± 0.27^b^	0.61 ± 0.56
T5	0.74 ± 2.98^b^	0.43 ± 0.43^b^	0.76 ± 1.58	3.99 ± 1.35^ab^	3.20 ± 3.01^b^	2.30 ± 0.36^b^	0.06 ± 0.29	0.37 ± 0.27^b^	0.31 ± 0.56
T6	14.17 ± 2.98^a^	2.72 ± 0.43^a^	1.88 ± 1.58	7.05 ± 1.35^a^	16.87 ± 3.01^a^	7.41 ± 0.36^a^	0.26 ± 0.29	6.93 ± 0.27^a^	2.08 ± 0.56
*p* value	0.015	0.01	0.10	0.01	0.01	<0.01	0.15	<0.01	0.13

**Table 4 toxins-17-00001-t004:** Effect of combined doses of mycotoxin on histopathological scoring on day 21. Histopathological changes for liver tissues were scored on 10 high-power randomly chosen fields/treatments as follows: no lesions (−), mild (+), moderate (++), and severe (+++).

Treatment	Hemorrhage	Hepatocyte Degeneration	Fibrosis in Portal Area	Bile Duct Proliferation
Control (T1)	−	−	−	−
T2	+++	+++	+++	+++
T3	++	++	++	++
T4	+++	+++	+++	+++
T5	+++	+++	++	++
T6	+++	+++	+++	+++

**Table 5 toxins-17-00001-t005:** Ingredient and nutrient composition of the basal diet (as-fed basis). Nutrients, vitamins, and minerals were provided in the form and amount described in the NRC, a standard reference diet for chickens [97].

Ingredient	Starter (%)
Corn	56.29
Soybean meal, 48% CP	37.87
Soybean oil	2.18
Dicalcium phosphate	1.48
Calcium carbonate	0.91
Sodium chloride	0.40
MHA	0.37
L-lysine	0.21
Trace mineral premix ^1^	0.10
Choline chloride (60%)	0.07
L-threonine	0.06
Vitamin premix ^2^	0.05
Phytase (500FTU)	0.01

^1^ Supplied per kilogram of diet: Mn, 107.2 mg; Zn, 85.6 mg; Mg, 21.44 mg; Fe, 21.04; Cu, 3.2 mg; I, 0.8 mg; Se, 0.32 mg. ^2^ Supplied per kilogram of diet: vitamin A, 5511 IU; vitamin D3, 1102 ICU; vitamin E, 11.02 IU; vitamin B12, 0.01 mg; biotin, 0.11 mg; menadione, 1.1 mg; thiamine, 2.21 mg; riboflavin, 4.41 mg; d-pantothenic acid, 11.02 mg; vitamin B6, 2.21 mg; niacin, 44.09 mg; folic acid, 0.55 mg; choline, 191.36 mg.

**Table 6 toxins-17-00001-t006:** Analyzed mycotoxin content of experimental diets. Representative samples of feeds from treatments (T) 1 to 6 were analyzed for FBs, deoxynivalenol (DON), and zearalenone (ZEA) concentrations by LC-MS/MS in Romer labs (Romer Labs, Union, MO, USA). The limits of detection (LOD) for mycotoxin analysis in feed samples, as performed by LC-MS/MS at Romer Labs, were as follows: aflatoxin B1-1.3 µg/kg, aflatoxin B2-1.2 µg/kg, aflatoxin G1-1.1 µg/kg, aflatoxin G2-1.6 µg/kg, and ochratoxin A-1.1 µg/kg. For acetyl DON, fusarenon-X, nivalenol, T-2 toxin, HT-2 toxin, neosolaniol, and diacetoxyscirpenol, the LOD was set at 100 µg/kg. Concentrations of the mycotoxins below the detection limits in the feed samples were not presented in the table. * Limit of detection.

Treatment	Total Fumonisins (FB1 + FB2 + FB3) (mg/kg)	FB1 (mg/kg)	DON (mg/kg)	ZEA (mg/kg)
T1 (Control)	0.8	0.6	0.4	LOD *
T2	3.6	2	2.5	0.9
T3	7.7	5	0.4	0.1
T4	14.0	9	3.5	0.7
T5	26.0	17	1.0	0.2
T6	33.0	21	3.0	1.0

**Table 7 toxins-17-00001-t007:** Primers sequence for qPCR. Primers were adopted from [98,99].

miRNA	Primer Sequence (5′-3′)	Annealing Temperature (°C)	miRBase Accession Number
gga-miR-29c-3p	CCAGACCCGGTAGCACCATTTG	58.5	MIMAT0001183
gga-miR-217-5p	CCACCATACTGCATCAGGAACT	58.5	MIMAT0001132
gga-miR-375	GCCGGATTTGTTCGTTCG	57.5	MI0003705
gga-miR-215-5p	GCCGCCATGACCTATGAAT	57.5	MIMAT0001134
gga-miR-155	CGGCGGTTAATGCTAATCGT	57.5	MI0001176
gga-miR-9-5p	ACGGCGGTCTTTGGTTATCTA	57.5	MIMAT0001195
gga-miR-19b-3p	GGCGGTGTGCAAATCCAT	57.5	MIMAT0001110
gga-miR-133a-3p	AACCCGTTGGTCCCCTTCA	58.5	MIMAT0001126
gga-let-7a-5p	GGCGGTGAGGTAGTAGGTTGT	60	MIMAT0001101
gga-U6	CTCGCTTCGGCAGCACA	60	GenBank accession no. NR004394
Universal Reverse Primer	Reverse primer for qPCR (Agilent High-Specificity miRNA QRT-PCR Detection Kit)	Proprietary sequence	

## Data Availability

The original contributions presented in this study are included in this article. Further inquiries can be directed to the corresponding author.

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
