# Peer review of "Early Biomarkers for Detecting Subclinical Exposure to Fumonisin B1, Deoxynivalenol, and Zearalenone in Broiler Chickens"

_toxins, 2024, doi:10.3390/toxins17010001_

Round 1

Reviewer 1 Report

Comments and Suggestions for Authors

This study investigates the effects of subclinical exposure to mycotoxins (FB1, DON, ZEA) on the physiological, biochemical, and molecular characteristics of broiler chickens and proposes potential biomarkers for the early detection of mycotoxin exposure. While the topic has certain practical significance, the study contains significant shortcomings in several aspects and fails to meet the standards for publication. The specific reasons are as follows:

1. The basal diet used in the study was naturally contaminated with a certain amount of FB1 and DON (0.4 mg/kg FB1 and 0.6 mg/kg DON). This background contamination significantly compromises the validity of the control group, making it difficult to attribute the observed effects to the mycotoxin exposure in the treatment groups rather than to the additional effects of the basal diet's inherent contamination.

2. Although the study included multiple dosage levels, it did not adequately discuss the clear relationship between mycotoxin concentrations and their effects. Furthermore, the rationale for selecting specific toxin concentration ratios was not explained, and the relationship between these concentrations and the FDA's recommended limits was insufficiently addressed. Additionally, the inconsistent variation in DON and ZEA levels across doses undermines the reliability of the dose-dependent analysis.

3. The study remains at the level of observing the correlations between miRNA expression and biochemical changes without delving into the specific mechanisms by which miRNAs are involved in mycotoxin-induced inflammation, apoptosis, or metabolic dysregulation. While the proposed biomarkers, such as serum AST, CK, FITC-d, gga-let-7a-5p, and gga-miR-217-5p, show statistical significance, these indicators are also subject to similar changes under other stress conditions. This lack of specificity limits their potential as unique markers for mycotoxin exposure.

4. The synergistic or antagonistic effects of multiple mycotoxin exposures should also be discussed.

5. The description of each parameter redundantly lists the mycotoxin concentrations for each treatment group, even though these treatment conditions were already clearly detailed in the experimental design section.

6. The histological results presented are based on observations from day 14, while the discussion focuses on histological changes on day 21. Additionally, the specific meanings of the three different colored arrow markers used in the figures are not explained.

7. The study aims to identify early biomarkers (at 14 days) for subclinical exposure to a combination of mycotoxins (FB1, DON, and ZEA). However, the discussion section primarily focuses on the analysis of data from day 21, overlooking the critical results related to early detection at day 14.

8. L126-129 and L150-152, the results described are inconsistent with the data shown in the figure. Table. 4, what are the evaluation criteria for +, ++, and +++? The biochemical parameters in Table 6 and Table 7 are inconsistent.

Author Response

Please see the below response:

Response to Reviewer 1 Comments

1. Summary

Thank you very much for taking the time to review this manuscript. Please find the detailed responses below and the corresponding changes in red color font in the re-submitted files.

2. Questions for General Evaluation

Reviewer’s Evaluation

Response and Revisions

Does the introduction provide sufficient background and include all relevant references?

Must be improved

Corrected

Are all the cited references relevant to the research?

Must be improved

Corrected

Is the research design appropriate?

Must be improved

Corrected

Are the methods adequately described?

Must be improved

Corrected

Are the results clearly presented?

Must be improved

Corrected

Are the conclusions supported by the results?

Must be improved

Corrected

3. Point-by-point response to Comments and Suggestions for Authors

Comments 1: The basal diet used in the study was naturally contaminated with a certain amount of FB1 and DON (0.4 mg/kg FB1 and 0.6 mg/kg DON). This background contamination significantly compromises the validity of the control group, making it difficult to attribute the observed effects to the mycotoxin exposure in the treatment groups rather than to the additional effects of the basal diet's inherent contamination.

Response 1: We appreciate the reviewer’s concern regarding the potential confounding effects of naturally occurring FB1 and DON in the basal diet. This is indeed an important consideration, and we would like to address it as follows: The natural contamination of feed with mycotoxins like FB1 and DON reflects real-world conditions. In commercial poultry house environments, poultry diets are very often contaminated with multiple mycotoxins. Even though we used clean corn to produce basal diet, natural contamination in the finished diet reflects the field conditions. The mycotoxin levels in the basal diet (FB1; 0.4 mg/kg and DON; 0.6 mg/kg) were much lower than the regulatory guidance levels (e.g., FDA limits: 50 mg/kg for FB1 and 5 mg/kg DON for broiler finished feed). Considering all experimental groups, including the control, were fed the same baseline diet, the background contamination had an identical impact on each group. This guarantees that variation observed between treatment groups can still be accounted to the additional mycotoxin exposure introduced in the treatments. By this way, a consistent baseline was established in this study. Further, the study still offers meaningful insights into how the combination of subclinical doses of mycotoxins impacts biochemical and molecular responses. These findings are particularly important for understanding real-world exposure conditions and can help in developing practical and effective strategies to mitigate mycotoxin-related risks in commercial settings.

These following sentences were included in the discussion section: “As mycotoxins are ubiquitous in nature, even though clean corn was used to produce basal diet, natural contamination in the finished diet reflects the field conditions. Hence, in this current study, the starter basal diet in the control group was naturally contaminated with 0.4 mg/kg FB1 and 0.6 mg/kg DON. The mycotoxin levels in the basal diet (FB1; 0.4 mg/kg and DON; 0.6 mg/kg) were much lower than the regulatory guidance levels (e.g., FDA limits: 50 mg/kg for FB1 and 5 mg/kg DON for broiler finished feed). Considering all experimental groups, including the control, were fed the same baseline diet, the background contamination had an identical impact on each group. This guarantees that variation observed between treatment groups can still be accounted to the additional mycotoxin exposure introduced in the treatments”.

Comments 2: Although the study included multiple dosage levels, it did not adequately discuss the clear relationship between mycotoxin concentrations and their effects. Furthermore, the rationale for selecting specific toxin concentration ratios was not explained, and the relationship between these concentrations and the FDA's recommended limits was insufficiently addressed. Additionally, the inconsistent variation in DON and ZEA levels across doses undermines the reliability of the dose-dependent analysis.

Response 2: We appreciate your comment regarding the relationship between mycotoxin concentrations, their effects, and the rationale behind the selected toxin concentration ratios. The following information was included in the discussion section.  

Addressing the variability in toxin concentrations among the treatment groups is that even though this study was aimed to achieve consistent toxin concentration, it was quite challenging with DON and ZEA culture materials. The most homogeneous distribution of culture material was achieved by thorough mixing before incorporation into the feed to simulate possible practical conditions. Despite thorough mixing, variability in toxin distribution ("hot spots") reflects the real-world problem of feed contamination, where toxins are rarely present in fixed ratios. In this current study, the mycotoxin concentration was chosen with the intention of simulating the real-world mycotoxin contamination scenarios that poultry producers very often see in commercial settings. These doses used in this experiment are in a range from low risk to high levels of subclinical exposures but do not exceed the FDA guidance limits. Our goal was aimed at examining the wide range of co-contamination patterns, including both sub-threshold and potentially harmful exposure levels, and the selected concentrations represent contamination levels frequently reported in field surveys.  Further, precise dose-response effects of mycotoxins are frequently complicated by so many factors, including environmental temperature and humidity. Hence, this study aimed to mimic natural contamination scenarios to better understand the effect of co-contamination of multiple mycotoxins on poultry health under practical conditions. The variability in toxin levels in culture material limiting precise dose-dependent analyses, but the observed trends were consistent across treatment groups, particularly at higher toxin concentrations, indicating a general dose-dependent relationship. Hence the doses we used in this study relevant for addressing real-world challenges faced by poultry producers and also providing initial insights about the risks associated with subclinical mycotoxin exposures.

Comments 3: The study remains at the level of observing the correlations between miRNA expression and biochemical changes without delving into the specific mechanisms by which miRNAs are involved in mycotoxin-induced inflammation, apoptosis, or metabolic dysregulation. While the proposed biomarkers, such as serum AST, CK, FITC-d, gga-let-7a-5p, and gga-miR-217-5p, show statistical significance, these indicators are also subject to similar changes under other stress conditions. This lack of specificity limits their potential as unique markers for mycotoxin exposure.

Response 3: We appreciate reviewer's comment. However, the primary goal of this study was to identify potential biomarkers, which is a first step for future investigations. We acknowledge that pathway-specific assays require more targeted approaches, which is beyond the scope of this work. We would like to point out that this present study serves as a critical starting point for those efforts. It is true that these biomarkers may react to general stressors however the study results are associated with mycotoxin exposure under controlled conditions. Furthermore, our result emphasizes the possible use of these markers in combination rather than separately, and the miRNA responses seen in the present study (e.g., gga-let-7a-5p, gga-miR-217-5p) are consistent with previous studies about these miRNAs response to inflammation and metabolic stress.  A biomarker panel that includes both miRNAs and biochemical indicators may enhance diagnostic specificity for mycotoxin exposure. We also note that while these biomarkers are not unique to mycotoxins, their consistent statistical significance under the tested conditions suggests their relevance as initial candidates for further validation.

Comments 4: The synergistic or antagonistic effects of multiple mycotoxin exposures should also be discussed.

Response 4: We appreciate the reviewer’s suggestion about the potential synergistic or antagonistic effects of multiple mycotoxins.

We agree that the interaction between mycotoxins is an important aspect of their impact on animal health. However, in this study, we did not have single mycotoxin treatment groups, as it aimed to simulate real-world contamination scenarios where co-occurrence of multiple mycotoxins is the norm. In commercial settings chickens are rarely exposed to individual mycotoxins. Therefore, the findings from this study provide insights into the combined effects of mycotoxins under practical conditions, even if precise interactions (synergistic, additive, or antagonistic) could not be separately evaluated. We acknowledge that single-toxin treatment groups would have allowed us to analyze specific interactions; however, outcomes vary depending on several factors including toxin type, dosage, and duration of exposure. For example, some studies have demonstrated that the co-occurrence of DON and ZEA has a synergistic effect on intestinal integrity (Jia et al., 2020; Ren et al., 2016; Thapa et al., 2021), while other combinations may show antagonistic effects (Jia et al., 2020; Lo et al., 2016). These interactions are variable and context-dependent, making it challenging to generalize results from isolated exposures. Despite the absence of single-toxin groups, the current study provides important information about the overall impact of naturally occurring multiple mycotoxins. These results are especially relevant to poultry where chickens are exposed to several mycotoxins.

The above information was included in the discussion section.

Comments 5: The description of each parameter redundantly lists the mycotoxin concentrations for each treatment group, even though these treatment conditions were already clearly detailed in the experimental design section.

Response 5: Thank you for pointing this out. We agree with this comment. Therefore, we have 

now detailed the description of treatments only in the materials and methodology section and we have deleted all the treatment names in the table and figure legends to avoid repetitive mentions of mycotoxin concentrations. In the discussion section, we recognize that some degree of repetition can aid readers who focus on isolated parts of the manuscript. To address this, we have streamlined the manuscript by removing redundant mentions of mycotoxin concentrations in the parameter descriptions to avoid unnecessary repetition while ensuring clarity for readers who may refer specific sections independently.

Comments 6: The histological results presented are based on observations from day 14, while the discussion focuses on histological changes on day 21. Additionally, the specific meanings of the three different colored arrow markers used in the figures are not explained.

Response 6: Apologies for not correcting this in the document. The histological results presented in the manuscript are of day 21.

·       The discrepancy in the description of histological results have been corrected accordingly in the text and in the table 4 and figure 5.

·       Under the figure 5 legend, the specific meanings of the three different colored arrow markers used in the figures are explained as “Fibrosis and bile duct proliferation (- (red)arrows) in portal areas and hepatocellular infiltration (- (black) arrows) and vacuole (- (green)arrows).”

Comments 7: The study aims to identify early biomarkers (at 14 days) for subclinical exposure to a combination of mycotoxins (FB1, DON, and ZEA). However, the discussion section primarily focuses on the analysis of data from day 21, overlooking the critical results related to early detection at day 14.

Response 7: Thank you for pointing this out. We agree with this comment. Therefore, we have updated our Discussion section to explain both the day 14 and day 21 results clearly. We have described the specific concentrations at which each biomarker can detect early mycotoxin exposure on day 14 and day 21. These combined doses of mycotoxins significantly upregulate miRNAs, suggesting that these miRNAs are particularly reliable for detecting mycotoxin exposure. While serum FITC-d, AST, and CK levels provide valuable information about mycotoxin-induced damage even at subclinical doses, changes in miRNA expression detected at day 14 suggest that miRNAs are more sensitive and timelier tools for identifying early disruptions in gene regulation. Further histological changes, such as necrosis, apoptosis, and gut integrity, were observed at day 21 and were preceded by alterations in miRNA expression at day 14. These results suggest that miRNA could serve as an early biomarker for subclinical mycotoxicity and a useful molecular tool for detecting mycotoxin exposure before severe pathological damage. This ability to detect changes in miRNA expression as early as 14-day exposure would help farmers take preventive measures and reduce economic losses

Comments 8: L126-129 and L150-152, the results described are inconsistent with the data shown in the figure. Table. 4, what are the evaluation criteria for +, ++, and +++? The biochemical parameters in Table 6 and Table 7 are inconsistent.

Response 8: Thank you for pointing this out. We agree with this comment. Therefore, we have rewritten the lines in the red color font.

·       The results in the lines 128-129 and lines 136-137 are modified and rewritten to be consistent with results shown in the figure as “On day 14, there were significant differences (P < 0.05) across treatment groups serum FITC-d concentrations when compared to the control group,” and “On day 21, there were significant differences (P < 0.05) across the treatment groups serum FITC-d concentrations when compared to the control group.”

·       The criteria for scoring +, ++, and +++ was based on histopathological parameters, including fibrosis in the portal area, bile duct proliferation, hepatocyte degeneration, and cellular infiltration, were observed under 20X magnification and scored as no changes (-), mild (+), moderate (++), and severe (+++), respectively, which is described in methodology section 5.8 under Histopathological analysis in lines 594-598.

·       Table 6 and 7 which are renamed as Table 2. Earlier the results for biochemical parameters total protein, calcium and potassium were not mentioned in d14 biochemical parameters as their values were outside the reference interval of Vetscan® VS2 Chemistry Analyzer which is used for analysis. Now, we have added those values in the d14 biochemical parameters table with a footnotes in lines 166-167.

5. Additional clarifications

All the corrections suggested by the reviewers have been rewritten or modified in the document in red color font.

Reviewer 2 Report

Comments and Suggestions for Authors

Many format aspects of this submission should be revised and corrected.

Abstract: Abbreviations for mycotoxins, although common (FB1, DON, ZEA), should be explained in the Abstract the first time they are used.

The same for FITC-d (explain all abbreviations at first use in abstract, text, tables and figures; there are many unexplained abbreviations in the present article)

Line 6: The use of the term ‘biomarkers of exposure’ can be confusing since in the present work it seems more appropriate to call them ‘biomarkers of effects’.

Lines 13 to 16: Were the significant differences observed in T6 birds on day14 also significant in relation to the control group?

Introduction

Lines 38-40: For comparison EU guidance values for mycotoxins in broilers are FB 20-60 mg/kg, DON 8-12 mg/kg and ZEA 2-3mg/kg (Commission Recommendation 2006/576/EC of 17 August 2006 on the presence of deoxynivalenol, zearalenone, ochratoxin A, T-2 and HT-2 and fumonisins in products intended for animal feeding)

Lines 43-45: The use of three decimal places tends to cause confusion between decimals and units of thousands even if the correct punctuation marks are used.

Line 39 and line 46: What is the subclinical dose of fumonisins in chickens 20 or 50 mg/kg?

Line 42: Please indicate the year of the Biomin survey (References 7 and 52 correspond to different years)

Results

How can it be that the first table in the article is Table 5? Anyway, Table 5 is missing the footnotes explaining the abbreviations and the statistical significance.

The same, how can it be that the first figure in the article is Figure 2?

Lines 121 to 129 and lines 139 to 146: The same paragraph is repeated twice

Lines 168-175: The same paragraph is repeated again

Lines 189 to 196: The same paragraph is repeated again

Lines 204 to 211: The same paragraph is repeated again

Line 217 to 224: The same paragraph is repeated again

Lines 237 to 234: The same paragraph is repeated again

Lines 267 to 274: The same paragraph is repeated again

Please homogenize the number of significant digits in all text, tables and figures.

Line 278: Tables 5, 6 and 7 show a single column with SEM values, but Table 8 shows SEM (or SD) values in each column. Please, homogenize.

Lines 290 to 298: The same paragraph is repeated again

Discussion

Line 311: Please indicate the year of the Biomin survey (References 7 and 52 correspond to different years)

Material and methods

Line 521: It is said that the limits of detection (LOD) for mycotoxin analysis in feed samples, were as follows: aflatoxin B1-1.3 ppb, aflatoxin B1-1.2 ppb (???). Also, do not use ppb, use µg/kg instead.

Author Response

Response to Reviewer 2 Comments

1. Summary

Thank you very much for taking the time to review this manuscript. Please find the detailed responses below and the corresponding revisions/corrections highlighted in red color font in the re-submitted files.

2. Questions for General Evaluation

Reviewer’s Evaluation

Response and Revisions

Does the introduction provide sufficient background and include all relevant references?

Can be improved

Corrected

Are all the cited references relevant to the research?

Can be improved

Corrected

Is the research design appropriate?

Can be improved

Corrected

Are the methods adequately described?

Can be improved

Corrected

Are the results clearly presented?

Can be improved

Corrected

Are the conclusions supported by the results?

Yes

3. Point-by-point response to Comments and Suggestions for Authors

Comments 1: Abstract: Abbreviations for mycotoxins, although common (FB1, DON, ZEA), should be explained in the Abstract the first time they are used.

The same for FITC-d (explain all abbreviations at first use in abstract, text, tables and figures; there are many unexplained abbreviations in the present article)

Line 6: The use of the term ‘biomarkers of exposure’ can be confusing since in the present work it seems more appropriate to call them ‘biomarkers of effects’.

Lines 13 to 16: Were the significant differences observed in T6 birds on day14 also significant in relation to the control group?

Response 1: Thank you for pointing this out. We have revised the abstract and corrected as suggested.

·       Explained the abbreviations for the first time for FB1, DON, ZEA and FITC-d in line 8-9. Also, we have explained the abbreviations for FDA, EU in the introduction in line 40-43 and BWG, FCR in the abstract and table footnotes (Table 1).

·       The term ‘biomarkers of exposure ’has changed to ‘biomarkers of effects’ in line 6.

·       There were no significant differences observed in bodyweight gain on day 14 in all the mycotoxin treatment groups in relation to the control group (P>0.05). This line has been added in the abstract in line 13-14.

Comments 2: Introduction

Lines 38-40: For comparison EU guidance values for mycotoxins in broilers are FB 20-60 mg/kg, DON 8-12 mg/kg and ZEA 2-3mg/kg (Commission Recommendation 2006/576/EC of 17 August 2006 on the presence of deoxynivalenol, zearalenone, ochratoxin A, T-2 and HT-2 and fumonisins in products intended for animal feeding)

Lines 43-45: The use of three decimal places tends to cause confusion between decimals and units of thousands even if the correct punctuation marks are used.

Line 39 and line 46: What is the subclinical dose of fumonisins in chickens 20 or 50 mg/kg?

Line 42: Please indicate the year of the Biomin survey (References 7 and 52 correspond to different years)

Response 2: Agreed with the reviewer. Corrected as suggested.

·       The EU guidelines with maximum permissible levels for FB, DON and ZEA are added in the manuscript along with reference provided as according to the European Union (EU) in lines 43-44, as “according to the European Union (EU), the maximum permissible levels for fumonisins (FB1+FB2), DON, ZEA in poultry feed are 20 mg/kg, 5mg/kg, and 0.25 mg/kg, respectively.”

·       The three decimal places in the line 43-45 are revised into single decimal as “The average concentration of FBs was 4.2 mg/kg, with a maximum of 83.3 mg/kg, while DON averaged 1.4 mg/kg, with a maximum concentration of 11.3 mg/kg detected in corn samples.”

·       Line 51-52 it is rewritten as “All the doses which fall below the FDA recommended maximum permissible levels are considered as subclinical doses, which is below 50 mg/kg for FB and 5 mg/kg for DON.”

·       The year of Biomin mycotoxin survey has been added and corrected for both the references as 2023 in the introduction (line 46) as well as in the discussion section (line 233).

Comments 3: Results

How can it be that the first table in the article is Table 5? Anyway, Table 5 is missing the footnotes explaining the abbreviations and the statistical significance.

The same, how can it be that the first figure in the article is Figure 2?

Lines 121 to 129 and lines 139 to 146: The same paragraph is repeated twice

Lines 168-175: The same paragraph is repeated again

Lines 189 to 196: The same paragraph is repeated again

Lines 204 to 211: The same paragraph is repeated again

Line 217 to 224: The same paragraph is repeated again

Lines 237 to 234: The same paragraph is repeated again

Lines 267 to 274: The same paragraph is repeated again

Please homogenize the number of significant digits in all text, tables and figures.

Line 278: Tables 5, 6 and 7 show a single column with SEM values, but Table 8 shows SEM (or SD) values in each column. Please, homogenize.

Lines 290 to 298: The same paragraph is repeated again

Response 3: Apologies for not correcting the table and figure numbers and thanks for the suggestions.

·       The table and figure numbers have been corrected as per the order throughout the document. Initially this difference in the table and figure numbers is because the materials and methodology section has been written below the introduction and while editing the document for submitting to the journal this section was moved to the end.

·       The repeated lines in the table and figure footnotes have been removed throughout the document as suggested.

·       The table and figures have changed to significant digits throughout the document.

·       Different numbers of SEM for the Table 3 (which was earlier Table 8) is because the there was one liver sample in T3 group having low RIN number, and it had only 5 replicates instead of 6 replicates for liver microRNA expression. This following sentence was included under methodology section. Line 554-555 “One sample in the T3 group with an RIN below 7.5 was excluded from cDNA synthesis. “

Comments 4: Discussion

Line 311: Please indicate the year of the Biomin survey (References 7 and 52 correspond to different years)

Material and methods

Line 521: It is said that the limits of detection (LOD) for mycotoxin analysis in feed samples, were as follows: aflatoxin B1-1.3 ppb, aflatoxin B1-1.2 ppb (???). Also, do not use ppb, use µg/kg instead.

Response 4: Agreed with reviewer comments. Corrected as suggested.

·       The year of Biomin mycotoxin survey has been added and corrected for both the references as 2023 in the introduction (line 46) as well as in the discussion section (line 233).

·       Apologies for the mistake. It was aflatoxin B1-1.3 ppb, aflatoxin B2-1.2 ppb and the ppb was replaced to µg/kg in the Table 6 footnote in the line 488-490 as “aflatoxin B1-1.3 µg/kg, aflatoxin B2-1.2 µg/kg, aflatoxin G1-1.1 µg/kg, aflatoxin G2-1.6 µg/kg, and ochratoxin A-1.1 µg/kg. For acetyl DON, fusarenon-X, nivalenol, T-2 toxin, HT-2 toxin, neosolaniol, and diacetoxyscirpenol, the LOD was set at 100 µg/kg.”

5. Additional clarifications

All the corrections suggested by the reviewers have been rewritten or modified in the document in red color font.

Round 2

Reviewer 1 Report

Comments and Suggestions for Authors
The authors have addressed all my concerns, I suggest that the manuscript can be accepted.

Reviewer 2 Report

Comments and Suggestions for Authors

The authors have responded appropriately to the questions and suggestions raised by the reviewer and, consequently, the manuscript has been satisfactorily reviewed and corrected.